# Double-duty caregivers enduring COVID-19 pandemic to endemic: "It's just wearing me down"

**Jasneet Parmar**[1], **Tanya L'Heureux**[1], **Michelle Lobchuk**[2], **Jamie Penner**[2], **Lesley Charles**[1], **Oona St. Amant**[3], **Catherine Ward-Griffin**[4], **Sharon Anderson**[1]*

1 Department of Family Medicine Faculty of Medicine & Dentistry, University of Alberta, Edmonton, Alberta, Canada, 2 Helen Glass College of Nursing, Rady Faculty of Health Sciences, University of Manitoba, Winnipeg, Manitoba, Canada, 3 Daphne Cockwell School of Nursing, Toronto Metropolitan University, Toronto, Ontario, Canada, 4 Arthur Labatt Family School of Nursing, University of Western Ontario, London, Ontario, Canada

* sdanders@ualberta.ca

**Data Availability Statement:** The data underlying the results presented in the study are available on Kaggle Double-Duty Caregivers. https://www.

## Abstract

The COVID-19 pandemic has considerably strained health care providers and family caregivers. Double-duty caregivers give unpaid care at home and are employed as care providers. This sequential mixed-method study, a survey followed by qualitative interviews, aimed to comprehensively understand the experiences of these Canadian double-duty caregivers amidst the pandemic and the transition to the endemic phase. The multi-section survey included standardized assessments such as the Double-duty Caregiver Scale and the State Anxiety Scale, along with demographic, employment-related, and care work questions. Data analysis employed descriptive and linear regression modeling statistics, and content analysis of the qualitative data. Out of the 415 respondents, the majority were female (92.5%) and married (77.3%), with 54.9% aged 35 to 54 years and 29.2% 55 to 64 years. 68.9% reported mental health decline over the past year, while 60.7% noted physical health deteriorated. 75.9% of participants self-rated their anxiety as moderate to high. The final regression model explained 36.8% of the variance in participants' anxiety levels. Factors contributing to lower anxiety included more personal supports, awareness of limits, younger age, and fewer weekly employment hours. Increased anxiety was linked to poorer self-rated health, and both perceptions and consequences of blurred boundaries. The eighteen interviewees highlighted the stress of managing additional work and home care during the pandemic. They highlighted the difficulty navigating systems and coordinating care. Double-duty caregivers form a significant portion of the healthcare workforce. Despite the spotlight on care and caregiving during the COVID-19 pandemic, the vital contributions and well-being of double-duty caregivers and family caregivers have remained unnoticed. Prioritizing their welfare is crucial for health systems as they make up the largest care workforce, particularly evident during the ongoing healthcare workforce shortage.

kaggle.com/datasets/sharonanderson/double-duty-caregivers/data.

**Funding:** Northern Alberta Academic Family Medicine Fund R16P06 (Dr. Jasneet Parmar). The funders had no role in study design, data collection and analysis, decision to publish, or preparation of the manuscript.

**Competing interests:** The authors have declared that no competing interests exist.

## Introduction

The COVID-19 pandemic has severely impacted healthcare providers, causing exhaustion, anxiety, and burnout [1–4]. In the early months of the pandemic, 26% of nurses worked overtime[5]. In 2021, approximately 20% of Canadian healthcare providers (236,000) worked overtime, averaging 8.2 hours/week of paid overtime and 5.8 hours/week of unpaid overtime [6]. Initially, healthcare providers were praised for their courageous care amid the risks, PPE shortages, and treatment uncertainties of COVID-19 [7]. However, as the pandemic continued into 2022, public gratitude diminished because of lockdown fatigue, vaccine conspiracy theories, and disparagement of medical knowledge [8]. Providers faced insults, threats, and increased stress as the pandemic persisted, alongside extended work hours, and changes in care delivery methods [6, 9, 10]. A survey conducted from September to November 2021 revealed that 92% of nurses, 83.7% of physicians, and 83.0% of personal support workers/healthcare aides and other providers felt more stressed at work compared to pre-pandemic times [11].

COVID-19 also presented a crisis for family caregivers [12–17]. Family caregivers are those who provide unpaid care for family, chosen family, friends, or neighbors with mental or physical illness, disabilities, or aging-related frailty. In the community, they handle 90% of care [18, 19], and assist with 15 to 30% of care in congregate care settings [20, 21]. The majority (84%) of family caregivers have a personal family connection to the care recipient, while 16% are not related [22]. The mental and physical health of care recipients deteriorated at an accelerated rate during the pandemic, [23] leading to increased caregiving hours and intensity [24, 25], amplifying the emotional and physical challenges of providing care [26, 27]. Family caregivers faced the dual concern of keeping their vulnerable loved ones and themselves safe from COVID-19, while also experiencing limited access to acute care, primary care, home care, and social care programs. They also had reduced opportunities for social activities and experienced higher levels of sleep disturbances, fatigue, anxiety, depression, and burnout [28, 29].

Family caregivers' care trajectories have become much longer, more demanding, and intense because of medical advancements, longer life spans, and neoliberal policies to reduce medical care costs but transfer care responsibilities onto family caregivers [30, 31]. Many family caregivers are now taking on medical and nursing tasks that were traditionally performed by regulated health professionals[32, 33]. These tasks include wound care, catheterization, managing medical equipment like pumps, intravenous, clysis, administering injections (including narcotics and anticoagulants), and providing tube feeding and medications [34, 35] In addition to these medical and nursing responsibilities, family caregivers also assist with extended and basic activities of daily living. They also bear the structural burden of care of navigating, negotiating, accessing, and coordinating resources within a siloed healthcare system [36–38].

Despite their essential role in healthcare, family caregivers remain under-engaged and under-supported within the healthcare system [31, 39] and their distress rates are rising [40, 41]. While most American sons or daughters caring for their parents (86%) reported giving care was a positive experience [42], distress rises as the care workload increases. The proportion of distressed caregivers increases dramatically for providing over 20 hours of care per week to a recipient who lives with them, has conditions like dementia, depression, responsive behaviors, or lives with severe disabilities [43]. Employed family caregivers may experience more anxiety. In the 2018 Statistics Canada General Social Survey-Caregiving (GSS-C), one in four Canadians of working age (19 to 70), totaling 5.2 million people, juggled family caregiving and paid employment [44, 45]. The hours dedicated to caregiving by both employed men and women increased by almost 50% between the 2012 and 2018 GSS-C surveys, with women going from 9.5 to 13.8 hours per week and men from 6.9 to 10 hours per week. Increased

caregiving hours raise the risk of work-family or family-work conflicts, where the demands or time requirements of work and family responsibilities clash [46, 47] and may contribute to stress and poor mental health [48, 49]. A recent study involving Canadian employees from various industries (automotive, manufacturing, insurance, research and development, health and social services, municipal government), highlighted that the number of hours spent organizing and coordinating healthcare tasks, rather than other family care tasks like housework, was associated with increased family-work conflict [50]. Employed family caregivers who dedicate over 20 hours per week to caregiving are 1.9 times more likely to experience poor work-life balance and twice as likely to leave their jobs compared to those providing less than 10 hours of care per week [45].

## Double-duty caregivers and the impacts of COVID-19

Double-duty caregivers are those who are employed in the healthcare field and provide unpaid care at home. While some of the double-duty caregiver literature includes unpaid care to dependent children [51–55], we limit our definition of double-duty caregivers to those in family caregiving roles. Catherine Ward-Griffin and colleagues theorized while employed family caregivers experience stress from balancing caregiving, work, and other life roles, double-duty caregivers face unique stress due to blurred boundaries between their professional and personal caregiving responsibilities [56, 57]. Recent work supports this theory [58].

A 2020 study focusing on registered nurses revealed the blurring of boundaries as nurses shifted between their professional and family caregiver roles [59]. They faced the challenge of deciding whether to disclose their professional identity to those caring for the person they themselves were caring for, all while managing their own expectations and those of staff and their families [57]. The study authors also reported that double-duty caregivers learned from their experiences in the health care system, which influenced their practices with patients and family caregivers, as well as their efforts to educate other professionals about the caregiving journey. Another study conducted in 2021 by Bristol and colleagues [60] reported that double-duty caregivers sought inclusion in the care provided, anticipating that their unique insider perspectives would be valued. Similar to the nurses in Ward-Griffin's original studies [61–63], nurses in Bristol's study also encountered a struggle between their professional and family caregiver roles. They were taken aback by healthcare professionals' expectation that they would accept care deficiencies [60]. This struggle intensified when interacting with other professionals who did not always appreciate their perspectives [60]. Similarly, critical care nurses who experienced a family member's illness also faced challenges in maintaining a balance between their professional persona and ensuring adequate care for their loved ones [64]. Many researchers studying double-duty caregivers propose that the additional emotional and care responsibilities of caring at home and at work leads to increased anxiety [54, 65, 66]. Recognizing the negative effects of the COVID-19 pandemic on family caregivers and healthcare providers, we wondered about the impacts of the COVID-19 pandemic on double-duty caregivers. In our literature search, we found a commentary by a medical student who was a family caregiver on their double-duty caregiver experience and the need for health care providers to be educated to communicate with family caregivers [65]. We found no other research on double-duty caregivers in the pandemic. This is a significant knowledge gap as at least one-third of employed health providers are also family caregivers [44].

## Methods

We conducted a sequential mixed-method study involving Canadian double-duty caregivers to capture the complexities of double-duty caregiving during the COVID-19 pandemic and

the transition to the endemic phase of the pandemic. The two-phase study approach, beginning with a quantitative survey followed by qualitative interviews, was conducted from April 21, 2022, to December 18, 2022. Given the diverse nature of family caregivers and their experiences, our aim was to gain a comprehensive understanding of their experiences during the COVID-19 pandemic and the shift toward the endemic phase. Endemic status is reached when a significant portion of the global population has been vaccinated or infected, and the virus continues to circulate at lower levels, akin to the seasonal flu [67]. The exact timing of the transition from the COVID-19 pandemic to the endemic phase is uncertain.

Building upon the theory put forth by Ward-Griffin and colleagues [39,40], we hypothesized that greater expectations and blurred boundaries would be associated with higher levels of anxiety, while increased support, awareness of limits, boundary-setting, and making connections would be linked to lower anxiety. We then hypothesized that caregiver health would be strongly associated with anxiety, and that age, gender, boundary blurring, expectations, support, awareness of limits, boundary-setting, connection-making, work hours, caregiving hours, and financial difficulties would also significantly contribute to caregiver anxiety. We also conducted interviews to further understand the experience of double-duty caregiving during the COVID-19 pandemic and its transition into an endemic phase.

To ensure the accuracy and relevance of our survey, our research team designed it and then had four double-duty caregivers from our team conduct a thorough pretest. Subsequently, we obtained approval from the ethics board to proceed with the study.

## Recruitment

We conducted an open-format survey using the secure REDCap [68–70] data collection platform from April 21 to June 30, 2022. To recruit participants, we utilized social media platforms such as LinkedIn, Twitter, Facebook, and Instagram, along with targeted emails to healthcare provider, disease-based and caregiver organizations containing recruitment material. Participants were asked three qualifying questions to be eligible for the survey: (1) "Do you look after someone (or help to look after someone) who has a disability, mental illness, drug or alcohol dependency, chronic condition, dementia, terminal or serious illness, needs care due to frailty and aging, and/or COVID-19?", and (2) "Are you a healthcare provider? By healthcare providers we mean any individual employed in health occupations who are working in health industries." Participants were required to read the ethics information and provided implied consent by clicking on the survey link (S1 File).

Out of the 606 individuals who clicked on the survey, 556 (81.17%) current double-duty caregivers responded to over three-quarters of the questions, and 415 (68.5%) completed the entire questionnaire. We included only surveys with complete data. Survey data were collected anonymously. Upon survey completion, participants were given the option to indicate their willingness to participate in a follow-up interview and/or receive notification about the study's results. The emails of those who agreed to these options were recorded separately on a REDCap survey so that identifying data could not be linked to the survey data.

Aligned with research advising that saturation is achieved with 12 to 15 interviews [71, 72], we sent emails to twenty respondents randomly selected from those who completed throughout the time the survey was active. All of our interview participants were female. We wanted to recruit interview participants by the three caregiving Interface groups, but our research ethics board wanted us to collect email addresses on a survey separate from data.

The research assistant arranged mutually agreeable interview times either via Zoom or with the first 18 respondents who provided contact information and who replied to the email. All were women. Because of the nature of our ethics (demographics not connected to identifying

data for interview participation) we could not recruit interview participants demographically or theoretically. The information letter and verbal consent form was emailed to participants. As per the University of Alberta Health Research Ethics Board ethics approval, the research coordinator reviewed the consent questions, and participants provided verbal consent at the outset of the interview.

## Data collection

The survey consisted of four sections: (1) standardized assessments, including the Double-duty Caregiver Scale [63], the six-item State Anxiety Scale [73], and a self-rated health assessment; (2) demographics related to the double-duty caregiver and care-receiver; (3) employment-related questions; and (4) questions pertaining to care work. The complete survey can be accessed in the Supporting Materials (S1 File, Survey in Word).

## Quantitative survey measures

Double-duty Caregiver Scale. Ward-Griffin and colleagues [63] developed the twenty-two-item Double-duty Caregiver Scale to investigate the experiences of healthcare providers who engage in simultaneous paid and unpaid caregiving, and its impact on their health and well-being. There are six dimensions in the Double-duty Caregiver Scale: Expectations, Supports, Knowing limits, Setting limits, Connections, and the Caregiving Interface. The Expectations domain (4 items) examines the expectations placed on double-duty caregivers from themselves, their families, and their profession. The Supports domain (4 items) assesses the availability of informational, emotional, and practical support from family, friends, professional colleagues, and the work environment. Knowing limits (2 items) explores double-duty caregivers' perceptions of their capacity to provide care, while Setting limits (2 items) captures the actions they take to maintain boundaries between their personal and professional caregiving domains. Connections (4 items) focuses on the strategies and actions that double-duty caregivers employ to access care and navigate the healthcare system within their family caregiving role. The Caregiving Interface domain (6 items) measures the extent to which boundaries between the professional and family care roles are blurred. Perceptions within the Caregiving Interface domain reflect double-duty caregivers' perceived ability to maintain boundaries, while consequences capture their emotional responses to role blurring and the psychological impacts thereof. Scores on the Caregiving Interface scale range from 6 to 30, with higher scores indicating greater blurring of boundaries and higher stress levels. In the original validation, the six subscales demonstrated satisfactory internal consistency with Cronbach's alpha coefficients exceeding.65. In this study, Cronbach's alphas exceeded.75. For a comparison of the Cronbach's alphas between the 2009 validation and the present study, please refer to S2 File, "Double-duty Caregiver Scale Dimensions."

Ward-Griffin and colleagues [63] identified three distinct experiences (prototypes) within the Caregiving interface: Making it work, Working to Manage, and Living on the edge. These experiences reflect the social processes and the degree of boundary blurring encountered by double-duty caregivers. The extent of boundary blurring is influenced by care expectations and the resources available to manage the demands. In the Making it work prototype, family caregivers perceive that they have the resources necessary to meet expectations, while those in the Living on the edge prototype experience significant boundary blurring without adequate resources to manage the high demands placed upon them. It is important to note that family caregivers can transition between these prototypes as expectations, demands, and resources evolve. The scores assigned to each prototype, as specified by the scale authors [55] are: 6–15 for Making it work, 16–21 for Working to Manage, and 22–30 for Living on the edge. For a

comprehensive explanation of the six dimensions, please refer to the Supporting Materials (S2 File, Double-duty Caregiver Scale dimensions).

### Six-Item State Anxiety Scale

We used the Six-Item State Anxiety Scale to assess anxiety [73]. This validated short-form scale, derived from the State-Trait Anxiety Inventory (STAI), measures feelings of apprehension, tension, nervousness, and worry. Participants responded to each item on a four-choice Likert scale, 1. Not at all, two. Somewhat, 3. Moderately, 4. Very Much Items 1, 3, and 6 were positively worded (with reverse scoring for absence of anxiety). To calculate the final score, we summed the scores for each item and multiplied the total by 20/6. STAI scores range from 20 to 80, with higher scores showing more severe symptoms. Research has shown that the six-item version is as reliable and valid as the original 20-item version [63,65,66], with reported Cronbach's alphas ranging from.74 to.82 [39]. In this survey, the Cronbach's alpha was.81. We categorized the STAI scores into two groups using cutoff scores: <40 for no or minimal symptoms, and ≥41 for moderate or severe symptoms [74–76].

### Double-Duty Caregiver's Health

To assess overall health, we employed a single question: "In general, would you say your health is . . .?" Participants could choose from response options such as poor, fair, good, very good, or excellent. This single question has been established as a reliable measure of health [77]. We also inquired whether participants perceived any changes in their physical and mental health over the past year, with response options including improved, remained about the same, deteriorated, or don't know/prefer not to answer.

Demographics included workplace setting (primary care, acute care, home care, supportive living, long-term care, social/community care), gender, age, ethnicity, urban/rural suburban, marital status, and educational level.

### Employment

We asked how many hours respondents were working in an average week with the responses of 1–14, 15–29, 30–34, 35–39, 40, 41–49 hours/week, 50 or more hours/week and don't know/ prefer not to answer.

### Family care work

The hours devoted to care work in an average week were assessed with the following answer options: < 9 h, 10–20 h, 21–39 h, and 40+ h. We also queried the duration of being a family caregiver, along with the number of family members, friends, or neighbors who provided help with family caregiving in the past 12 months. Information about the individuals receiving care, such as age, relationship, and living situation, was also requested.

### Financial stress

We asked about impact of caregiving on their financial situation with the question, "In the last year during the COVID-19 pandemic, have you experienced financial hardship because of your caregiving responsibilities?" with the responses of No, A few, Moderate, and A lot of financial hardships because of my caregiving responsibilities.

## Qualitative data collection

We collected qualitative data in the survey and in-depth qualitative interviews. In the survey, we included three open-ended qualitative questions to gather more in-depth insights from participants. These questions were: (1) How has your employment affected your family caregiving? (2) How has caregiving affected your employment? and (3) Is there anything you would like to share about double-duty caregiving or any suggestions for supporting double-duty caregivers?

For the interviews, we developed a semi-structured interview guide divided into three sections. Firstly, we aimed to establish rapport with the participants and encourage their perspectives on how caregiving and employment had changed or remained the same during the pandemic. We asked about their experiences in both caregiving and employment roles. Then, we delved into whether any of the three Double-duty Caregiver Prototypes (Making it work, Working to manage, Living on the edge) represented their experience and if they had transitioned from one prototype to another, exploring the reasons behind these shifts. The semi-structured interview guide can be found in S3 File.

## Data analysis

Quantitative Survey Data. We utilized descriptive statistics, such as mean, standard deviation, frequency, and proportion, to summarize the demographic characteristics of the participants. To assess the reliability of the standardized scales, we calculated internal consistency measures. Then we conducted linear regression models to explore the associations between anxiety and the factors of the Double-duty Caregiver Scale. Next, we performed exploratory regression models to investigate the relationships between anxiety and caregiver demographic characteristics, Double-duty Caregiver Scale factors, self-rated health, weekly employment hours, weekly work hours, and financial distress. We retained factors with a significance level of .06 as sometimes insignificant factors can be significant in more complex linear model [78, 79] and subsequently conducted three hierarchical linear regression models to assess the unique associations of these factors with anxiety. The quantitative analyzes were conducted using SPSS Statistics Version 27 (IBM®, Chicago, IL, USA) [80].

## Qualitative Interview and Survey Data

In the qualitative analysis we explored the themes related to the experiences of double-duty caregivers, including expectations, support, knowing and setting limits, making connections, and the caregiver interface (degree of blurring between employment and unpaid care roles). The interviews were transcribed verbatim, and the research assistant and coordinator carefully listened to the recordings to correct any transcription errors. We removed the identifying information from the transcripts to ensure anonymity.

We employed Hsieh and Shannon's [81] qualitative summative content analysis methods to analyze the data from the interviews and qualitative survey responses. First, the cleaned transcripts were imported into the NVivo® (QSR International) qualitative analysis program, to facilitate data management. Then, our analysis involved inductively identifying exemplars of the components derived from the theoretical framework of double-duty caregiving. We did this by thematically coding the underlying meaning of words and sentences. Throughout the analysis, we compared and contrasted participants' perspectives on each of Ward-Griffin's theoretical components of the double-duty caregiving experience. After completing the thematic coding, we conducted word searches for each component and conducted a deductive analysis based on the results. Then we compared our deductive analysis with our inductive themes and interpretations to ensure the credibility and internal consistency of the inductive analysis [81].

We followed Thorne's recommendations by checking the transcripts for accuracy, utilizing memos, and maintaining an audit trail to enhance the trustworthiness and rigor of our qualitative analysis [82, 83].

## Results

### Participant characteristics

Of the 415 people who completed the survey over half (54.9%) were 35 to 54 years of age and 29.2% were 55 to 64. The majority identified as female (92.5%) and married (77.3%). They identified 47 different occupational roles. Just over a third (34.5%) were nurses, followed by diagnostic imaging specialists (14.7%), social workers/ mental health councillors (7%), radiation therapists/ technologists (5.3%) and physicians (4.6%). Over half lived in urban areas (58.8%) with equal proportions from suburban (20.7%) and rural areas (20.7%). Just over a third (38.1%) had dependent children. At the time of the survey, almost half (42%) reported working in paid employment 40 or more hours weekly and 17.3% were employed 29 hours or less. Over two-thirds (68.9%) estimated their mental health had deteriorated in the past year, and 60.7% indicated that their physical health worsened. A third (34%) rated their health as fair or poor, 40.5% as good, 20.5% as very good, and 4.8% as excellent. See Table 1 Demographics.

They had been providing unpaid care for a mean of 8.5 years (median 6 years) and almost half (47%) for parents, a quarter (25.3%) for children with special needs, and 13.7% for a spouse. The mean age of the person they cared for was 59.3 years, ranging from an infant born with disabilities to a 100-year-old adult. Most care recipients (84.6%) lived in community homes, with 46% living with the double-duty caregiver and 38.6% living in a separate home. There were equal proportions living in supportive living (7.7%) and long-term care (7.7%). Thus, most (80.7%) reported a travel time of 30 minutes or less to their receiver's residence. Just over half (54.4%) cared for 20 hours or less, 21.7% for 21 to 40 hours, and 21.2% for 41 or more hours weekly. When we divided participants with complete scores into three prototypical groups using Ward-Griffin's guidance for Double-duty Caregiver Interface scores, 5.4% were Making it work, 28.7% were Working to manage, and 65.8% were Living on the edge at the time of the survey. Aligned with the high proportions who felt they were Living on the edge, the mean on the Six-Item Anxiety Scale was 50.9 (SD 15.5, Median 50). Scores of $\geq$41 indicate moderate to severe anxiety. Three-quarters (75.9%) of the double-duty caregivers' self-rated anxiety was moderate to high.

As we could expect with higher anxiety, the proportions of those Living on the edge (65.8%) were higher in this sample than Ward-Griffin's 2014 population (53.1%) [56, 57]. The 2022 Making it work(5.4%) and Working to Manage (28.7%) prototype proportions were lower than Ward-Griffin's [56, 57] 12.5% and 34.4% respectively.

### Double-duty caregiver scale factors associated with anxiety

Results from the hierarchical linear regression model testing the associations between the factors in the Double-duty Caregiver Scale are summarized in Table 2. There were no violations of assumptions of normality, linearity, multicollinearity, and homoscedasticity. The total variance explained by the model was 29.4%. Hypothesis one that the Double-duty Caregiver Scale factors would significantly predict anxiety was partially supported. The statistically significant factors (in order of beta values) were Caregiving Interface perceptions (β = .202), supports-personal(β = -.208), Caregiving Interface- consequences (β = .179), knowing limits (β = -.125) and setting limits (β = -.096).

**Table 1. Demographics, and key caregiving characteristics (n = 415).**

| | N (%) |
|---|---|
| Age | 415 |
| 15–34 years | 40 (9.6) |
| 35–54 years | 228 (54.9) |
| 55–64 years | 121 (29.2) |
| 65–74 years | 25 (6.0) |
| Prefer not to answer | 1 (0.2) |
| Gender | N (%) |
| Male | 28(67) |
| Female | 384 (92.5) |
| LGBTQ2 | 1(0.002) |
| Prefer not to answer | 2 (0.005) |
| Marital Status | |
| Married/common law | 321 (77.3) |
| Widowed | 7 (1.7) |
| Separated/ Divorced | 44 (10.6) |
| Single never married | 36 (8.7) |
| Prefer not to answer | 7 (1.7) |
| Education | N (%) |
| Some college or less | 9 (2.2) |
| Certificate | 22 (5.3) |
| Diploma | 120 (28.9) |
| Bachelor's degree | 163 (39.3) |
| Graduate degree | 91 (21.9) |
| Prefer not to answer | 10 (2.4) |
| Profession | N (%) |
| Nurse (Nurse Practitioner, Geriatric, Registered, Licensed Practical, Psychiatric) | 143 (34.5) |
| Diagnostic Imaging specialist | 61(14.7) |
| Social worker/ mental health councillor | 29 (7.0) |
| Radiation therapist/ Radiation Technologist | 22 (5.3) |
| Physician | 19 (4.6) |
| Health care aide/ Personal Support Worker | 16 (3.9) |
| Paramedic, EMT | 14 (3.4) |
| Pharmacist/ Pharmacy Technician | 13 (3.1) |
| Dietician | 9 (2.2) |
| Speech-Language Therapist | 6 (1.4) |
| Assistants, Activity, Therapy, Medical | 6 (1.4) |
| Recreation Therapist | 5 (1.2) |
| Physical Therapist | 4 (1.0) |
| Psychologist | 4 (0.7) |
| Respiratory Therapist | 3 (0.7) |
| Other | 61 (14.7) |
| Has Dependent Children | N (%) |
| Yes | 158 (38.1) |
| No | 239 (57.6) |
| Prefer not to answer | 18 (4.3) |
| Double-Duty Caregiver Residence Location | |
| Urban | 243 (58.5) |

*(Continued)*

**Table 1.** (Continued)

| | N (%) |
|---|---|
| Suburban | 86 (20.7) |
| Rural | 86 (20.7) |
| Prefer not to answer | 0 (0) |
| Years providing care for family member, chosen family, friend or neighbor (n = 404) | |
| Mean | 8.5 y |
| Median | 6.0 y |
| Ethnicity | N (%) |
| White Caucasian | 331 (79.9) |
| Filipino | 15 (3.6) |
| Canadian | 14 (3.4) |
| Black | 4 (1.0) |
| First Nations | 6 (1.4) |
| Hispanic or Latinx | 1(0.2) |
| Metis | 9 (2.2) |
| South Asian | 8 (1.9) |
| Southeast Asian | 9 (2.2) |
| West Asian | 4 (1.0) |
| Other | 5 (1.2) |
| Prefer not to answer | 9 (2.2) |
| Employment Setting | N (%) |
| Primary Care | 66 (15.9%) |
| Emergency Medical Care | 21 (5.1%) |
| Acute Care | 128 (30.8) |
| Home Care | 28 (6.7%) |
| Supportive/ Assisted Living | 12 (2.9%) |
| Long-term Care | 29 (7.0%) |
| Community/ Social Care | 130 (31.3) |
| Prefer not to answer | 1 (0.2) |
| Relationship to person cared for | N (%) |
| Parent/ parent-in-law | 195 (47.0) |
| Spouse/partner | 57 (13.7) |
| Child | 105 (25.3) |
| Chosen Family | 10 (2.4) |
| Sibling | 14 (3.4) |
| Other Relative | 19 (4.6) |
| Friend | 4 (0.9) |
| Neighbour | 1 (0.2) |
| Employer/Employee | 4 (0.9) |
| Prefer not to answer | 6 (1.5) |
| Age of Person Cared for | N (%) |
| Range | Less than 1 y to 100 y |
| Mean (SD) | 59.3 (29.9) |
| Median | 70.5 |
| Residence of Person you spend most time caring | |
| Same home (live with) | 191 (46.0) |
| Separately (Private home, condo, apartment | 160 (38.6) |
| Supportive or assisted living | 32 (7.7) |

(*Continued*)

**Table 1.** (Continued)

| | N (%) |
|---|---|
| Long-term care | 32 (7.7) |
| Prefer not to answer | 0 (0) |
| Travel Time to Person spent most care time | N (%) |
| Lives with | 191 (46.0%) |
| ≤ 30 Min | 144 (34.7%) |
| 31–59 Min | 40 (9.6%) |
| 1–2 Hours | 18 (4.3%) |
| 2.1–6 hours | 4 (1.0%) |
| 6+ hours | 15 (3.6%) |
| Prefer not to answer | 3(0.7%) |
| Has your employment status changed due to family caregiving? | |
| Yes | 81 (19.5) |
| No | 326 (78.6) |
| Prefer not to answer | 8 (1.9) |
| Financial Hardships due to caregiving | |
| No financial hardships | 190 (45.8) |
| A few financial hardships | 120 (28.9) |
| Moderate financial hardships | 59 (14.2) |
| A lot of financial hardships | 45 (10.8) |
| Prefer not to answer | 1 (0.2% |
| Changes to Caregivers' physical health in last year | |
| Improved<br>Remained stable | 28 (6.7)<br>125 (30.1) |
| Deteriorated | 252 (60.7) |
| Prefer not to answer | 10 (2.4) |
| Changes to Caregivers' mental health in last year | |
| Improved<br>Remained stable | 20 (4.8)<br>102 (24.6) |
| Deteriorated | 286 (68.9) |
| Prefer not to answer | 7 (1.7) |
| Weekly hours of employment | |
| Up to 29 h | 72 (17.3) |
| 30–39 h | 169 (40.7) |
| 40–49 h | 140 (33.7) |
| 50 + h | 34 (8.2) |
| Prefer not to answer | 0 |
| Weekly Family caregiving hours | |
| ≤10 h | 159 (38.3) |
| 11–20 h | 67 (16.1) |
| 21–30 h | 58 14.0) |
| 31–40 h | 32 (7.7) |
| 41–80 h | 44 (10.8) |
| ≥ 81 h | 43 (10.4) |
| Prefer not to answer | 12 (2.9) |
| Six-Item Anxiety Scale | |
| Range | 20–80 |
| Mean (SD) | 50.9 (15.5) |

(*Continued*)

**Table 1.** (Continued)

|  | N (%) |
|---|---|
| Median | 50.0 |
| Low Anxiety ≤ 41 | 98 (23.6) |
| Moderate to High Anxiety ≥ 42 | 315 (75.9) |
| Preferred not to answer 1+ questions | 2 (.05) |
| Self-Rated Health | N (%) |
| Excellent | 20 (4.8) |
| Very good | 85(20.5) |
| Good | 168 (40.5) |
| Fair | 116 (28.0) |
| Poor | 25 (6.0) |
| Prefer not to answer | 1(.2) |
| Changes in Physical Health in last year | N (%) |
| Improved | 28 (6.7) |
| Remained the same | 125 (30.1) |
| Deteriorated | 252 (60.7) |
| Prefer not to answer | 10 (2.4) |
| Changes in Mental Health in last year | N (%) |
| Improved | 20 (4.9) |
| Remained the same | 102 (24.6) |
| Deteriorated | 286 (68.9) |
| Prefer not to answer | 7 (1.7) |
| Double-Duty Caregiving Scale | Mean (SD) |
| Expectations Familial (Range 2–10) | 7.58 (2.04) |
| Expectations Professional (Range 2–10) | 9.08 (1.35) |
| Expectations Total (Range 4–20) | 16.66 (2.92) |
| Supports Personal (Range 2–10) | 7.22 (1.86) |
| Supports Professional (Range 2–10) | 5.48 (2.08) |
| Supports Total (Range 4–20) | 12.72 (3.34) |
| Knowing Limits (Range 2–10) | 7.43 (2.01) |
| Setting Limits (Range 2–10) | 5.58 (1.92) |
| Caregiving Interface Perceptions (3–15) | 13.14 (3.74) |
| Caregiving Interface Consequences (3–15) | 10.58 (2.74) |
| Caregiver Interface Total (6–30) | 23.01 (4.55) |
| Caregiving Interface Scale into Theoretical Categories | |
| Making it work (Scores 6–15) | 22 (5.3%) |
| Working to manage (Scores 16–21) | 116 (28.0) |
| Living on the edge (Scores 22–30) | 266 (64.1) |
| Preferred not to answer one or more questions | 11 (2.7) |

Our second regression analysis regressed anxiety on the demographic factors and predictor variables. In Step 1, with the demographic variables of age and gender, the model was significant (F (2, 374) = 6.07, $p$ = .004, $R^2$ = .003). In Step 2 with double-duty caregiving variables, the model was also significant (F (7, 371) = 20.35, $p$ = < .001) accounting for 30% of the variance ($R^2$ = .300). Younger age, more personal supports, and knowing limits were significant in reducing distress. Both the Caregiving Interface perceptions and consequences of the blurring of boundaries between family caregiving and employment were significant in increasing distress. In Step 3, adding time spent in family care work and employment, perceptions of

**Table 2. Linear regression results of the double-duty caregiver scale factors.**

| | Standardized Coefficients | t | p Sig. | 95.0% CI for B | |
|---|---|---|---|---|---|
| | Beta | | | Lower | Upper |
| (Constant) | | 9.004 | < **.001** | 40.272 | 62.778 |
| Expectations Professional | 0.036 | 0.674 | **0.501** | -0.637 | 1.301 |
| Expectations Familial | -0.043 | -0.819 | **0.413** | -0.9 | 0.371 |
| Supports Professional | -0.074 | -1.432 | 0.153 | -1.079 | 0.17 |
| Supports Personal | -0.208 | -4.02 | < **.001** | -2.12 | -0.727 |
| Knowing Limits | -0.125 | -2.698 | **0.007** | -1.352 | -0.212 |
| Setting Limits | -0.096 | -2.087 | **0.038** | -1.202 | -0.036 |
| Making Connections | -0.027 | -0.579 | 0.563 | -0.395 | 0.216 |
| Care interface Perceptions | 0.202 | 3.588 | < **.001** | 0.467 | 1.6 |
| Care Interface Consequences | 0.179 | 3.215 | **0.001** | 0.322 | 1.337 |

R2 = .294, F = 17.110 [9, 378], p = < .001

financial distress from caregiving, and self-rated health significantly improved the model (Δ R$^2$.096, F change (4, 367) = 13.52 $p$ = < .001). This final model explained 36.8% of the variance. Hypothesis two that caregiver health would be strongly associated with anxiety, and that age, gender, boundary blurring, expectations, support, awareness of limits, boundary-setting, connection-making, work hours, caregiving hours, and financial difficulties would also significantly contribute to caregiver anxiety was partially supported. Personal supports (standardized β = -.16), knowing limits (standardized β = -.13), younger age (standardized β = -0.11), and fewer weekly employment hours (standardized β = -0.09) and were significant factors in reduced anxiety. Poor health (standardized β = .28), blurring boundaries, both perceptions (standardized β = .19) and consequences (standardized β = .12) were significant factors in increased anxiety (See Table 3). Gender, familial and professional expectations, professional supports, making connections, caregiving hours and financial difficulties were not significant factors in anxiety.

## Qualitative results

In what follows first, we provide a composite overview of the double-duty caregivers experiences. Then we turn to the themes related to the domains of the Double-duty Caregiver scale. Inductive and then deductive analysis of the eighteen interviews and qualitative questions in the survey revealed a more holistic understanding of how the COVID-19 affected double-duty caregivers' lives. Overall, the COVID-19 pandemic increased stress in both double-duty caregivers' employment and care roles. While workplace shifts varied at the beginning of the pandemic, participants all noted that their stress increased as hospitals tried to discharge patients earlier to make room for the COVID-19 patients. Employment workloads exploded with additional patients, resource shortages, and staff illness. Being seconded to work in another setting added to employee stress. Care work at home ballooned as home care services, community programs, and schools reduced services or closed. Those caring for community dwelling care receivers reported it was difficult to find staff to provide in-home care or that home care staffing was not reliable. Congregate care closures to family caregivers and visitors were distressing to these double-duty caregivers who were used to caring and advocating for care.

Participants' descriptions portrayed moral distress at the quality of care and support provided to patients at work and the difficulty in arranging for and/or providing care at home. Moral distress arises when one knows the right thing to do, but institutional constraints make

**Table 3. Hierarchical regression analysis predicting anxiety.**

| | F (df) | $R^2$ | Adj. $R^2$ | $\Delta R^2$ Change | F Change | Standardized B | t | p | Lower CI | Upper CI |
|---|---|---|---|---|---|---|---|---|---|---|
| **Step 1** | **378** | **0.031** | **0.026** | **.031** | **6.072**\*\* | | | | | |
| **Age** | | | | | | **-0.17** | **-3.43** | **< .001** | **-0.93** | **-0.25** |
| Gender | | | | | | -0.03 | -0.59 | 0.56 | -1.32 | 0.71 |
| **Step 2** | **371** | **0.300** | **0.283** | **0.269** | **20.357**\* | | | | | |
| **Age** | | | | | | **-0.14** | **-3.08** | **.002** | **-0.76** | **-0.17** |
| Gender | | | | | | 0.20 | 0.45 | .65 | -0.68 | 1.09 |
| Supports Professional | | | | | | -0.06 | -1.15 | .25 | -0.29 | 0.08 |
| **Supports Personal** | | | | | | **-0.22** | **-4.46** | **< .001** | **-0.64** | **-0.25** |
| **Knowing Limits** | | | | | | **-0.11** | **-2.29** | **.02** | **-0.37** | **-0.28** |
| Setting Limits | | | | | | -0.08 | -1.83 | .07 | -0.33 | 0.01 |
| Making Connections | | | | | | -.011 | -0.24 | .81 | -0.10 | 0.08 |
| **Care Interface Perceptions** | | | | | | **0.20** | **3.86** | **< .001** | **0.16** | **0.48** |
| **Care Interface Consequences** | | | | | | **0.16** | **2.98** | **.003** | **0.08** | **0.37** |
| **Step 3** | **390** | **0.368** | **0.09** | **0.096** | **13.529**\* | | | | | |
| **Age** | | | | | | **-0.11** | **-2.60** | **0.01** | **-0.66** | **-0.09** |
| Gender | | | | | | 0.03 | 0.71 | 0.48 | -0.43 | 1.13 |
| Supports Professional | | | | | | -0.02 | -0.49 | .62 | -0.22 | 0.13 |
| **Supports Personal** | | | | | | **-0.16** | **-3.31** | **.001** | **-0.50** | **-0.12** |
| **Knowing Limits** | | | | | | **-0.13** | **-3.09** | **0.00** | **-1.31** | **-0.29** |
| Setting Limits | | | | | | -0.06 | -1.38 | 0.17 | -0.28 | 0.05 |
| Making Connections | | | | | | -0.04 | -0.32 | .75 | -0.10 | 0.07 |
| **Care Interface Perceptions** | | | | | | **0.19** | **3.74** | **< .001** | **0.14** | **0.45** |
| **Care Interface Consequences** | | | | | | **0.12** | **2.30** | **0.02** | **0.02** | **0.30** |
| Financial difficulty none- a lot | | | | | | 0.08 | 1.69 | 0.09 | -0.72 | 0.94 |
| **Weekly employment hours** | | | | | | **-0.09** | **-2.08** | **0.39** | **-0.39** | **-0.01** |
| Weekly care work hours | | | | | | 0.02 | 0.49 | 0.62 | -0.11 | 0.18 |
| **Self-rated health excellent to poor** | | | | | | **0.28** | **6.29** | **< .001** | **0.76** | **1.45** |

\*\* $p = .003$ \* $p = < .001$

it nearly impossible to pursue the right course of action [84]. Almost all found it difficult to achieve a work-life balance, noting that self-care suffered. As the pandemic dragged on and moved to the endemic phase, participants described how ongoing staff shortages at work and at home continued. The difficulty navigating the health system, Navigating: "if I didn't have a health care background, I could not do this" was an overarching theme throughout the interviews and the double-duty caregiver domains. The themes in the Double-Duty Caregiver domains include:

1. Expectations: "I'm overwhelmed and I'm exhausted" with subthemes in Self Expectations: "It's just part of who I am." and Familial expectations "Basically the perfect ingredients."

2. Supports: "It's like juggling balls" with subtheme on Making Connections "I had support because I worked in a hospital."

3. The Caregiving Interface has three subthemes Making it work: "some sense of normality." Working to manage "It's a fine line" Living on the edge: "I was barely holding it together."

**Navigating: "If I didn't have a health care background, I could not do this.".** In both the survey and interviews, participants shared instances of encountering complex and confusing processes and rules within the healthcare, education, and legal systems. In the interviews, participants expressed astonishment at their own struggles in understanding how to access relevant information and resources, leading them to wonder how individuals without insider knowledge managed. The existence of silos and lack of coordination further compounded the challenges they faced, resulting in hurdles and delays when attempting to address issues or receive necessary support. Many participants expressed frustration over the lack of and unavailability of information on the options or resources that were available. They were surprised when healthcare providers failed to effectively communicate about the options and resources.

The navigation difficulties were further exacerbated by the COVID-19 pandemic. Participants stressed the urgent need for comprehensive support for individuals lacking specialized system knowledge and professional connections. They strongly recommended the provision of guidance and support throughout the navigation process, delivered by a reliable person or resource capable of offering information clearly, addressing questions, and assisting family caregivers in making informed decisions within the health and social care systems. Throughout the different phases of the COVID-19 pandemic, including before, during, and in the endemic stages, friends, neighbors, and family members turned to these double-duty caregivers for information and support in navigating the complex health and social care systems.

*And if I didn't have a health care background I could not do this. I feel really bad for parents who have to navigate the system and don't know the system. . . .I feel like there's a huge gap, that was even greater in the pandemic. #13*

*Of all the stages, finding supports and navigation for older people is horrendous. So just trying to figure out home care, okay, do I need a referral? Well, what are the rules, I couldn't even figure it out. Like I'm online, I'm going through within the system, even outside of the system, I can't figure it out. I can't figure out how to get into decent care and figure out what care he's getting. And then not knowing what other options there are. And you hear about something, but things aren't published. Right? #5*

**Expectations: "I'm overwhelmed and I'm exhausted".** Participants' own, their families, and their employment expectations collided in COVID-19. Participants stressed that before COVID, they faced immense pressure as double-duty caregivers. Then the COVID-19 pandemic exacerbated the expectations at home and at work, leaving many participants exhausted. They mentioned making sacrifices and their efforts to manage expectations and protect their families going unnoticed.

*The expectations to manage your family and that of others was exhausting before the pandemic. Now, there is no relief. You cannot complain about what others do—career limiting even if it is valid. So you go unnoticed, cleaning up errors in order to protect your family. [Survey]*

*I have to be honest. I'm overwhelmed and I'm exhausted. It is in you to help and caregive so you try to accept extra shifts and help where it is needed, however you are needed at home too. I think one of the hardest things is still the expectations of [disability program], home care, and [self-managed care] is so overwhelming. . . ... I feel like I am in a wheel did the exhaustion of caretaking come first? or did the exhaustion of my work come first? I have been a Nurse for*

*30 years, I have never had these feelings before. I believe the start of these feelings came during the 1st wave of the pandemic, and they have not subsided yet. #11*

**Self expectations: "It's just part of who I am.".** During the interviews, the role of being a caregiver emerged as a fundamental aspect of their identity. Participants strongly identified themselves as family caregivers, and their self expectations centered around a profound sense of responsibility and commitment. They expressed a deep-rooted belief in providing care to their family and friends, both at home and at work, with one participant stating, "I think it's just part of who I am." Their commitment to delivering high-quality care was unwavering, regardless of the challenges they faced. As professionals, they knew what good care should be.

Despite the increased demands placed on them due to the COVID-19 pandemic, they felt it was even more imperative to continue caring for their loved ones. The care-receiver's well-being was a primary concern, and they were worried about the potential impacts of COVID-19. While recognizing the importance of balancing their caregiving role with other aspects of their lives, including work, they emphasized the continued dedication to providing care despite the added responsibilities brought on by the pandemic.

*So I have to say that I think I've always had that [caring for others] in me. Like I've always looked after things for people and stuff like that. I think it's just part of who I am. But it has been crazy since COVID. #18*

*And I'm a nurse and I've been working full time to extreme overtime hours and then being the only medical person and the only girl child in my family, every time something's going on, I'm leaving work to go and take care of one of them, either of them, both of them. #2*

**Familial expectations "Basically the perfect ingredients.".** One participant used "the perfect ingredients" to capture the cultural or personal values influencing the familial expectations in her caregiving decisions. Subthemes familial, personal, and cultural values of love, reciprocity, and duty to family were the ingredients woven through the "natural" or familial care expectations. The majority reported that responsibility for caregiving fell directly or indirectly on them because they were the member of the family with knowledge in the medical field. More than half reported that family members did not understand how the healthcare system operated, thus the family expected them to step in. The risk of COVID-19 to frail or immunocompromised family members increased expectations for their care expertise and health care provider vigilance.

*Expectations of care provided by me to my family member have significantly increased since the pandemic started. I had a dramatic mental health crisis at work that resulted in some improvements both personally and professionally, but overall I still have a great amount of daily pressure to be a fulltime caregiver for my family member while being 110% at work and constantly willing to work extra shifts, overtime, and "continually improve daily". There is only so much one person can do. I also experienced unresolved COVID-19 related trauma that I have spent many, many hours trying to resolve in a satisfactory manner. [Survey]*

*I have heard that family caregivers in this fight experience high expectations that are difficult to manage and have difficulty accessing resources. I think that would be me. I am the daughter and I am in health care so I am expected to care, expected to know, but I still can't get the care that I think she needs. #16*

Healthcare employment expectations: "There is no end to work expectations except to 'do more'". Participants reported that their anxiety increased as the demands at work increased in

COVID. They spoke about lack of resources, constant changes in work obligations, heightened professional expectations to take extra shifts/work overtime, and difficulty managing work-life balance. About a quarter of the interview and survey participants highlighted the unfair expectations of women, especially nurses, to be resilient and manage multiple responsibilities without giving them time to care for themselves. That pressure increased exponentially in the pandemic and has continued with ongoing staff shortages.

*Work is crazy and makes it very challenging to manage work-life balance, to attend my caregiver responsibilities. There is no end to work expectations except to 'do more' but within regular work hours as overtime is not honored. If you say "no", you're not a team player; if you ask for help, you're 'struggling', if you work outside of work hours, you 'exhibit poor time management'. It's extremely stressful, creating constant sense of anxiety, I've become very introverted and quiet so I'm not asked to 'do more'; I don't want to be on my manager's radar, because she always has more 'opportunities' she wants addressed. Very unsatisfying work environment. #3*

**Supports: "I*t's like juggling balls".***   Double-duty caregivers' experiences with support from their workplace and community resources varied significantly depending on their situation. Some reported having strong support systems in place, with understanding managers, supportive coworkers, and/or access to community resources. Others received scant support from their workplace, families, and/or communities. In COVID, even strong support systems were challenged. They highlighted the shift of care burden from external sources to themselves, which led several to request time off work during COVID-19. Initially closures of schools, caregiver support services, such as adult day centers, respite care, and in-person support groups, non-essential businesses, and service providers increased their care responsibilities.

As the COVID pandemic dragged on and was declared endemic, staff burnout and shortages meant scrambling for people to provide care. Both at work and home, these double-duty caregivers were asking for scarce supports. In the absence of workplace policies to support family caregivers, at work, they were asking immediate supervisors or coworkers for accommodations, such as time off or flexible schedules to manage their caregiving. Then at home they were asking family, friends, and health and social care providers for help to organize care. Aligned with the double-duty caregiving theory, a third of interview participants spoke directly to professional expectations that they could, and should, provide care at home because they were qualified health professionals. They expressed how this led to reduced supports.

*Another impact of COVID is the fact that I can work from home. It's given me 3 hours a day between not commuting, getting ready and that kind of stuff. So that's increased my capacity to care at home. Home care basically said you've got the qualifications, so you do it. And I get it. They're strapped, too. And I have a friend whose husband's in-home care and the fact that she lives with you, she's triaged low because they triage the seniors that are by themselves higher up. So, because I'm here and have the flexibility within my day to try to add it in, I can do it, but like my best friend says, it is like juggling balls. Like you can only have so many balls before you're starting to drop balls. And I am definitely dropping balls for sure and have been for actually quite some time. #3*

**Knowing limits/ setting limits: "Employers see us as the Florence Nightingale skills".**
Overall, these quotes shed light on the complexities family caregivers faced and the significance of establishing and maintaining boundaries in their roles. That complexity increased in COVID as both caregiving and employment demands increased. Participants discussed the

challenges of managing expectations, setting clear boundaries, and finding balance amid multiple roles and responsibilities in each of those roles.

It was challenging to negotiate boundaries in complex family dynamics and care demands and increased demands at work. In the interviews, the double-duty caregivers emphasized the importance of recognizing personal limits, seeking support, and advocating for healthier work environments. However, they also noted that they didn't set limits until they were under extreme stress, "Living on the edge." Often, discussions about boundaries included notions of pushing back on expectations or advocating for better working conditions in response to ongoing pressure.

> *I'm giving up the extreme expectations. Employers see us as the Florence Nightingale skills and like those Amazon people, they think that we're just going to give, give, give, give to this career. But there has to be a point where enough. Like, I'm going to start pushing you back now because that's enough. Look, the expectations are just too much. We need boundaries on those expectations. #1*

**Making connections "I had support because I worked in a hospital".** Double-duty caregivers used their healthcare training and connections in their care setting to support their family caregiving. Even in the pandemic, they understood the importance of being assertive to ensure their loved ones' needs were met. Their insider knowledge allowed them to access support systems within the healthcare setting, seek advice and recommendations from social workers, pastoral care staff, and other professionals in various healthcare settings. Their healthcare training and experience equipped them to effectively seek answers, help, and referrals to improve the quality of care for their family members.

However, with their healthcare training, they often observed and experienced challenges within the healthcare system, citing instances of poor care, lack of attention, or neglect by healthcare professionals. Some family caregivers were critical of nurses and other healthcare workers whom they believed prioritized financial incentives, such as overtime pay, over providing quality care, which could have negative consequences for patients.

> *I feel that I had support because I worked in a hospital and there's a social worker there and the pastoral care guy and you talk to other people you know within the hospital. And if I ever had any question about what needed to be done, I could always talk to somebody, and they would give a recommendation. And then I would act on that with care for my mom especially. #7*

**The caregiving interface.** While in Ward-Griffin's [62, 63] theoretical model, the Caregiving Interface includes both the double-duty caregivers' perceptions of their ability to maintain care boundaries and the emotional or psychological consequences of the blurring of the boundaries, our participants described enormous challenges of being overwhelmed with work. Besides the stress of contracting COVID-19 at work and potentially infecting the vulnerable individuals they cared for at home, they described the emotional and psychological stress of trying to manage increased demands at work and home and the moral distress of not being able to get or give the quality of care they expected from their training and values. Participants' descriptions of moving from one prototype to another illustrated how slight changes could reduce or increase their anxiety.

**Making it work: "Some sense of normality.".** In this prototype, expectations are low and support resources are robust. Interview participants who reported they were now making it work expressed relief from uncertainty, worry, and anxiety; being content with their caregiving situation; and experiencing "some sense of normality" or work-life balance. Their workload

and stress were reduced by changing jobs, reducing hours or leaving work, being able to work from home; have dependable, quality help with caregiving at home; consistent support from family, friends, or colleagues; flexibility at work; or the care work changed because the care-receiver was placed in care, their condition improved, or they passed away.

> *No, I mean, I think I'm definitely, like, Making it work. Unfortunately, my dad's not here, so I don't have his caregiving. Personally, I am doing well. I have less anxiety and stress because I don't have those aspects of caregiving in my life. If I was still caregiving now, I think it would still be Working to manage because even though the world has opened up, the facilities haven't necessarily. I think that the restrictions would still make it challenging to provide the quality of life that we would want for my dad. #13*

**Working to manage "It's a fine line".** In this prototype, double-duty caregivers described the struggle to find a balance between their job demands and care work at home as a "fine line". Stress, anxiety, and difficulty maintaining their own physical health increased. It was a struggle to find resources to manage their caregiving responsibilities. Many were frustrated with the lack of support from the care providers they employed, home care services, family members, and/or workplace accommodations. About a quarter reported having to make compromises in their professional lives to manage their caregiving demands. They changed jobs, asked for leave of absences, turned down or didn't apply for promotions, or dropped to part-time or resigned.

> *You can go from it's a crisis all the time to oh, it's okay now. And I think I try to live as balanced as I can. Is it always perfect? Well, no. Right now my mom's in the hospital because she broke her pelvis. When I'm not working, I'm normally at the hospital. But I also can't afford to get sick. So that is a very fine line, right? So Working to manage now, but I could easily be Living on the edge. #4*

> *At the beginning of COVID I was probably at the Making it work stage, but now I'm Working to manage. It's just wearing me down. Yes, it started with the worry and the stress that came from and still comes from isolation and having to stay away from my parents and now even more so because their condition has deteriorated. #15*

**Living on the edge: "I was barely holding it together.".** In this prototype, workloads and expectations are exceedingly high, while resources are scant. All, but one caregiver we interviewed, acknowledged "Living on the edge" at some point after the COVID-19 pandemic began. Barely holding it together, moral distress over care or not meeting the quality of care they expected to provide either at home or work, or both were the threads running through participants' descriptions of the Living on the edge prototype.

> *I was definitely Living on the edge the whole entire time. And as more people came down with COVID at work, like those weeks that I was working like 60 hours a week because they couldn't get anybody else. Professionally in the workplace, I would say I was in the middle, but we didn't have a ton of cable. We just had a lot of stuff to manage. It was okay. But at home, I felt like it was having a bigger effect. I really wasn't able to maintain my responsibilities, so I definitely struggled more with home. It's been rough for me mentally. And I've just not wanted to seek out a lot of help. Like I went to the doctor, and he put me on meds for PTSD. I tried to come off them because I don't like meds. But yeah. . .. [long pause], I definitely had some days where I just felt like I was barely holding it together. I am still hanging on by my fingernails. #1*

*So, I felt like I was in fight or flight 24 hours a day, whether I was at work or at home or with my parents. My job changed because I had to navigate with the mandates and other things were different. I just did whatever I had to do to make it through the day. And honestly, like probably six months to nine months of my last few months in that role, I have a very low recollection of the work I did because I was in survival mode. #12*

## Discussion

First, we hypothesized that greater expectations and blurred boundaries would be associated with higher levels of anxiety, while increased support, awareness of limits, boundary-setting, and making connections would be linked to lower anxiety. It was partially supported. We found that both perceptions and consequences of blurred boundaries (Caregiving Interface) were significantly associated with higher anxiety. Personal supports and knowing and setting limits were significant in reducing anxiety. Second, when examining factors beyond the double-duty caregiving scale, we hypothesized that better caregiver health, younger age, male gender, less boundary blurring, fewer expectations, greater support, more awareness of limits, more connections, fewer weekly work and caregiving hours, and fewer financial difficulties would be predictive of less caregiver anxiety. Hypothesis two was also partially supported. In the final regression model, better health, personal supports, knowing limits, younger age, fewer weekly employment hours and setting limits were significant factors in reduced anxiety. Blurring boundaries, both perceptions and consequences, and financial difficulties from caregiving were significant factors in increased anxiety. Neither familial nor professional expectations nor employment supports, setting limits, and making connections were significant contributors to the model. We now discuss these quantitative results, considering our qualitative findings and the literature.

The qualitative data consistently highlighted the overarching theme of navigation difficulties within health and community systems. Family caregivers often find themselves becoming the de facto navigators in these systems [10,11,49,50]. Before the pandemic (2017), Taylor and Quesnel-Vallée estimated that family caregivers spend a significant portion of their time (15 to 50%) on the structural burden of care, which involves assessing and understanding needs, advocating for care, accessing services, and coordinating various aspects [37]. In their study of double-duty caregivers, St. Amant and colleagues [47] identified similar tasks, such as assessing, advising, advocating, collaborating, coordinating, and consulting associated with the professionalization of family care.

Notably, our study participants wondered how family caregivers coordinate and provide care without formal healthcare training. Similar observations have been made by Valaitis et al. [85] and Funk [86, 87] who reported that health and social care systems are challenging to navigate because of restrictive eligibility criteria, complex application processes, and other gatekeeping mechanisms. Besides complex navigation, family caregivers are expected to understand and manage complex health information and perform intricate medical tasks [31, 88]. Even prior to the pandemic, over half of family caregivers were already performing medical tasks typically handled by licensed healthcare professionals, including wound care, administering fluids and medication, and operating medical equipment such as health monitors and ventilators [17, 31, 89]. There is a pressing need for well-coordinated, integrated health and social care systems that support collaborative partnerships with patients and family caregivers [85, 86].

The quotes about expectations, "*It's just part of who I am,*" "*Basically the perfect ingredients*" and "*There is no end to work expectations except to 'do more'*" exemplified the heightened expectations double-duty caregivers experienced at work and at home since the pandemic

began in March 2020. They felt a strong sense of responsibility, committing to caregiving both at home and in their professional roles. Professional expectations included increased work demands, lack of resources, and challenges in maintaining work-life balance. Expectations stemmed from cultural values of women and those with medical knowledge being expected to provide care and health providers as self-less, "projected to care like Florence Nightingale". Some participants highlighted the unfair burden placed on women in healthcare, expecting "Florence Nightingale skills" expecting them to be resilient and manage multiple responsibilities without allowing time for self-care. These pressures intensified during the pandemic, exacerbated by ongoing staff shortages. Balancing employment, family caregiving, and personal life became increasingly challenging, leading to heightened anxiety. Most paid care workers in Canada are women (75% in 2016) [90]. During the COVID-19 pandemic, employment evolved differently for men and women in care occupations. While both men and women in care occupations experienced employment losses, women suffered greater losses than men (62.5%) at the beginning of the pandemic [91]. The gender differences lessened but continued October 2020 to February 2021 with women accounting for 56.4% of the year-over-year employment losses. The authors speculate that gendered care responsibilities may have caused women to stay at home.

Feminist scholar, Tronto argues that the traditional gender script in our society has embedded caring as women's work [92]. She suggests women are expected to care for their families, neighbors, and friends, and that they do so by doing the direct work of caring. Tronto has long charged that culturally, care is undervalued, invisible, underpaid, and without adequate support [93–96]. She contends traditionally we have viewed care as the private work of families, rather than a public responsibility. Care roles and responsibilities are unequal and unbalanced. Privileged groups and men can absolve themselves of the responsibility for care. The result is a lack of support for both paid care workers and unpaid family carers who are primary women.

Notably, in June 2020 as they called for papers about the impact of the COVID-19 pandemic, Tronto and Fine suggested the pandemic had highlighted the importance of care and the need for greater public support for care workers and family carers [97]. By 2022 however, they were more pessimistic, suggesting a return to the pre-COVID cultural status quo [10]. They argued we should culturally recognize care as a social justice issue; then emphasized the importance of transforming toward a caring society based on democratic care practice, where all voices and perspectives are included and heard. With the ongoing shortage of health care providers and the importance of work-life balance to well-being, there is an urgent need to reflect on traditional expectations of care and to move forward with a new ethic of caring that is grounded in justice, equity, and diversity, thereby creating a more inclusive and equitable culture of care that values the well-being of the people providing care.

Personal but not professional supports or using professional connections to obtain care were significant factors in reducing double-duty caregiver's anxiety. In the interviews, the double-duty caregivers leveraged their knowledge and expertise as health professionals to navigate the health care system and access support and care for the people they cared for. In fact, they marveled at how family caregivers, without their healthcare knowledge, managed. Even by using their healthcare experience to access supports, personal and professional support resources varied significantly. Family caregivers and their care situations are diverse and change along the care trajectory [98–101]. Health providers should take a proactive approach in assessing double-duty caregivers' skills, knowledge, and needs and offer **all** family caregivers the guidance, education, and resources needed to support them in their roles. Policy makers and researchers could engage double-duty caregivers to co-design better systems to support family caregivers and the people they care for.

Caregiving roles at home need to be acknowledged and supported with health system workplace policies. Participants described ad hoc situations that depended upon immediate leadership. Several of the double-duty caregivers we interviewed were circumspect about revealing their caregiving role to colleagues or leadership at work. In the interviews, two cited preferring to be private for not revealing they were family caregivers at home. Others were concerned their caregiving responsibilities might be perceived as distraction or that it might affect their ability to meet work demands. Comments in the survey and interviews also highlighted the lack of workplace policies to support family caregivers. All the double-duty caregivers in the interviews were aware of employee assistance programs, but only one reported it helped.

Flexible work schedules, especially flexibility to manage the care-receiver's appointments or crisis were valued. A recent review reported that because of the 1993 United States Family Medical Leave Act to support work-life balance, some American employers are offering paid emergency adult care days plus subsidized in-home adult care to accommodate the unpredictable nature of adult caregiving (41.25% of the workplaces in the review) [102]. However, the lower cost support groups, counseling and workshops were more frequently offered. Many of our participants noted employers or leadership were unable or unwilling to provide flexible work arrangements. Survey and interview respondents commonly reported being forced to take holidays or sick days to accommodate their caregiving. Interview respondents stressed it was difficult to get regular in-home staff during the pandemic and staffing shortages continue to make arranging alternative care challenging. Researchers report that immediate managers or leaders play a critical role in recognizing employee stress and deciding how to allocate support or caregiving policies [70–72]. Whether employees ask for supports or leaders apply caregiver support policies depends on the workplace culture [72]. Caregiving employees are more likely supported in workplace cultures that view work-life balance or caregiver-friendly workplace polices as an important to their business practices [72].

Survey and interview participants were particularly disturbed by the inequity related to differences in pay scales in hospitals, home care, and long-term care for providers with the same qualifications or staff hired temporarily through agencies and exclusions for certain groups of healthcare workers for pandemic payments. They attributed some of the difficulty getting home care and home care staffing shortages to higher pay scales in acute care than in long-term care and then in-home care. In their 2013 policy brief, Ward-Griffin and colleagues recommended enhancing community and home care system supports for family caregivers juggling employment and caregiving. The research on caregiver-friendly workplace policies is also relevant to health care systems [103, 104]. Since 2013, the landscape family caregivers navigate is increasingly challenging, marked by prolonged and demanding care trajectories because of advancements in medicine, longer lifespans, and cost-cutting measures that shift care responsibilities onto families [4, 50, 58]. These caregivers now shoulder not only personal care and extended activities of daily living, but also medical and nursing duties traditionally done by regulated health professionals. In our siloed health and social care systems, they are the de facto care coordinators finding, negotiating for, and coordinating resources [38, 105].

Setting limits and knowing limits were both significant factors in reducing double-duty caregiver distress. In the survey responses and interviews, participants reported trying to maintain a clear boundary between their personal and professional lives. However, setting limits became increasing more difficult as employment and/or caregiving demands increased. Increased time spent caring at work and at home reduced double-duty caregivers' family, social, and self-care. They had fewer opportunities for activities that bring joy in life such as connecting with family and friends or engaging in exercise or personally meaningful activities. Thus, it is not surprising that blurring boundaries, both perceptions and consequences were significant factors in increased anxiety. As we could expect with higher anxiety, the

proportions of those Living on the edge (65.8%) were higher in this sample than Ward-Griffin's 2014 [56, 57] population (53.1%). The 2022 Making it work (5.4%) and Working to Manage (28.7%) prototype proportions were lower than Ward-Griffin's [56, 57], 12.5% and 34.4% respectively. Notably, 60.7% perceived their physical health declined in the last year and 68.9% reported their medical health worsened. It is critical to improve the working conditions for double-duty caregivers and working caregivers.

The strain on the healthcare workforce and system was building long before COVID-19 began. In 1995, Celia Davies pointed out the unsustainable workload for nurses, management's struggle to cope, and a disconnect in leadership [106]. Adding to these challenges, from 2005 to 2018 front-line healthcare providers have experienced increased job insecurity and fewer support systems as governments tried to reduce or contain health care costs [107]. Currently, Canada is facing an unprecedented shortage of health care professionals [108]. The health and social services sector vacancy rate was 5.7% in November 2022, down from a multi-year high of 6.6% in September 2022 [109]. Burnout from the COVID-19 pandemic and demand from treatment postponed during the pandemic contributed, but an aging workforce, increased demands from an older population, providers overwhelmed by non-medical demands, and younger professionals who are working fewer hours to achieve a better work-life balance contribute to the provider shortage that is not expected to ease for at least 6 years [108].

Based on their research, Ward-Griffin and colleagues [110] called on health care administrators, human resource managers, researchers, policy makers, health provider associations, and union officials to work collaboratively to enhance workplace supports and human resource policies that recognize and support double-duty caregivers and create caregiver-friendly workplaces. They invited national, provincial, and territorial professional associations from across Canada to lobby governments to institute caregiver rights that recognize the value of family caregivers' unpaid labor. Very little has been done. The Canadian Centre for Caregiving Excellence charges the patchwork of provincial, territorial, and federal caregiving policies is failing [4]. The health and social care systems that family caregivers, healthcare providers and care recipients rely on for support have been overextended after years of under-funding, and now because of the impacts of the COVID-19-pandemic.

By 2046, a striking 20% of Canadian—the aging baby boomers and their echo boomer children—will be 65 years or older [111]. Despite the dedicated efforts of healthcare staff, Canada's health and continuing care systems are unprepared for the mounting demand. We do not have 10 years to make the changes that the system needs. Next year, the baby boomers turn 75. If we persist with the status quo, the operating costs of continuing care will double by 2032 [112, 113]. Innovative solutions are imperative in tackling this looming care crisis head-on.

## Strengths and limitations

A strength of this study is use of validated scales, including the Double-duty Caregiver Scale to examine double-duty caregivers' experiences quantitatively and qualitatively. It highlights the anxiety related to the unique expectations from overlapping employment and family care roles. The cross-sectional snapshot is a limitation. While it captured increased stress at work and home, longitudinal research is needed. We know very little about how double-duty caregiving affects work careers longitudinally.

## Conclusion

The shortage of healthcare workforce should prioritize the concern of double-duty caregivers. The COVID-19 pandemic has shed light on various care issues, but attention has waned. Family caregiving, which was previously invisible, remained unnoticed during the pandemic as

well. Double-duty caregiving was even more invisible. However, the care needs of the Canadian population are increasing as the proportion of older adults increase. Given that family caregivers form the largest care workforce, that includes the subset know as double-duty caregivers, it is crucial for health systems involved in care to prioritize their double-duty caregiver workforce and family caregivers' well-being. Researchers, policymakers, administrators, human resource managers, health provider associations, and citizens need to reevaluate the ethics and equity surrounding caregiving and double-duty caregiving.

## Supporting information

**S1 File. Survey questions.**
(PDF)

**S2 File. Double-duty caregiver scale dimensions.**
(DOCX)

**S3 File. Semi-structured interview guide.**
(DOCX)

## Acknowledgments

Thank you to the survey and interview participants who gave generously of their time.

## Author Contributions

**Conceptualization:** Jasneet Parmar, Tanya L'Heureux, Michelle Lobchuk, Jamie Penner, Lesley Charles, Oona St. Amant, Catherine Ward-Griffin, Sharon Anderson.

**Data curation:** Jasneet Parmar, Tanya L'Heureux, Michelle Lobchuk, Jamie Penner, Lesley Charles, Oona St. Amant, Catherine Ward-Griffin, Sharon Anderson.

**Formal analysis:** Jasneet Parmar, Tanya L'Heureux, Michelle Lobchuk, Jamie Penner, Lesley Charles, Oona St. Amant, Catherine Ward-Griffin, Sharon Anderson.

**Funding acquisition:** Jasneet Parmar, Michelle Lobchuk, Jamie Penner, Lesley Charles, Sharon Anderson.

**Investigation:** Jasneet Parmar, Tanya L'Heureux, Michelle Lobchuk, Jamie Penner, Lesley Charles, Oona St. Amant, Catherine Ward-Griffin, Sharon Anderson.

**Methodology:** Jasneet Parmar, Tanya L'Heureux, Michelle Lobchuk, Jamie Penner, Lesley Charles, Oona St. Amant, Catherine Ward-Griffin, Sharon Anderson.

**Project administration:** Jasneet Parmar, Tanya L'Heureux, Michelle Lobchuk, Jamie Penner, Lesley Charles, Sharon Anderson.

**Resources:** Jasneet Parmar, Tanya L'Heureux, Michelle Lobchuk, Jamie Penner, Lesley Charles, Sharon Anderson.

**Software:** Jasneet Parmar, Tanya L'Heureux, Michelle Lobchuk, Lesley Charles, Sharon Anderson.

**Supervision:** Jasneet Parmar, Tanya L'Heureux, Michelle Lobchuk, Lesley Charles.

**Validation:** Jasneet Parmar, Tanya L'Heureux, Michelle Lobchuk, Oona St. Amant, Catherine Ward-Griffin, Sharon Anderson.

**Visualization:** Tanya L'Heureux, Michelle Lobchuk, Oona St. Amant, Sharon Anderson.

**Writing – original draft:** Jasneet Parmar, Tanya L'Heureux, Michelle Lobchuk, Jamie Penner, Lesley Charles, Oona St. Amant, Catherine Ward-Griffin, Sharon Anderson.

**Writing – review & editing:** Jasneet Parmar, Tanya L'Heureux, Michelle Lobchuk, Jamie Penner, Lesley Charles, Oona St. Amant, Catherine Ward-Griffin, Sharon Anderson.

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
