## [Decision Letter · Decision Letter 0]

16 Jan 2024

PONE-D-23-31596Double-Duty Caregivers Enduring COVID-19 Pandemic to Endemic: “It’s just wearing me down”PLOS ONE

Dear Dr. Anderson,

Thank you for submitting your manuscript to PLOS ONE. After careful consideration, we feel that it has merit but does not fully meet PLOS ONE’s publication criteria as it currently stands. Therefore, we invite you to submit a revised version of the manuscript that addresses the points raised during the review process.

Below you will find the reviewers' comments. Please address the reviewers' comments so that after adequate response I can further consider your article for publication. Also note that reviewer #1 has provided comments in attachment, while the comments of reviewer #2 are mentioned further in the body of this mail.

We look forward to receiving your revised manuscript.

Kind regards,

Stefaan Six, Ph.D.

Academic Editor

PLOS ONE

2. In the ethics statement in the Methods, you have specified that verbal consent was obtained. Please provide additional details regarding how this consent was documented and witnessed, and state whether this was approved by the IRB.

“Thank you to the survey and interview participants who gave generously of their time.

This study was funded by the Northern Alberta Family Medicine Fund”

“Northern Alberta Academic Family Medicine Fund”

6. In the online submission form you indicate that your data is not available for proprietary reasons and have provided a contact point for accessing this data. Please note that your current contact point is a co-author on this manuscript. According to our Data Policy, the contact point must not be an author on the manuscript and must be an institutional contact, ideally not an individual. Please revise your data statement to a non-author institutional point of contact, such as a data access or ethics committee, and send this to us via return email. Please also include contact information for the third party organization, and please include the full citation of where the data can be found.

7. We notice that your supplementary tables are included in the manuscript file. Please remove them and upload them with the file type 'Supporting Information'. Please ensure that each Supporting Information file has a legend listed in the manuscript after the references list.

Reviewers' comments:

Reviewer's Responses to Questions

**Comments to the Author**

1. Is the manuscript technically sound, and do the data support the conclusions?

Reviewer #1: Yes

Reviewer #2: Yes

2. Has the statistical analysis been performed appropriately and rigorously? 

Reviewer #1: Yes

Reviewer #2: Yes

3. Have the authors made all data underlying the findings in their manuscript fully available?

Reviewer #1: Yes

Reviewer #2: Yes

4. Is the manuscript presented in an intelligible fashion and written in standard English?

Reviewer #1: Yes

Reviewer #2: Yes

5. Review Comments to the Author

Reviewer #1: this manuscript is exceptionally well written and highly comprehensive. The authors related effectively a period that was detrimental to caregivers. their design is rigorous and reviewing it was a real delight.

I added minor comments to enhance the flow but generally speaking this manuscript has 5 stars.

Reviewer #2: Dear author

Your article concerning double-duty caregivers’ experiences during the COVID-19 pandemic is very interesting and relevant. The comments below are in support or clarification of some issues regarding the methodology.

Kind regards

Comments:

1. 202-204: How did you deal with the questionnaires that were not fully completed? Were they included/excluded?

2. 208: Can you explain why you chose 18 respondents for the follow-up interviews? + Can you provide the participants’ characteristics (how many of them were female/male/x)…?

3. 258: Can you provide all four options of the Likert scale to the reader? + Why did you choose for a four-point Likert scale? For example, to avoid neutral answers?

4. 292: How many of the participants answered the open-ended qualitative questions?

5. 322-323: Can you explain why you chose the qualitative summative context analysis methods of Hsieh and Shannon?

6. 319-321: Did you obtain data saturation?

In general: Please place the references before the full stop that marks the end of the sentence.

6. PLOS authors have the option to publish the peer review history of their article (what does this mean?). If published, this will include your full peer review and any attached files.

Reviewer #1: No

Reviewer #2: No

---

## [Author Response · Author response to Decision Letter 0]

25 Jan 2024

https://journals.plos.org/plosone/s/file?id=wjVg/PLOSOne_formatting_sample_main_body.pdf andhttps://journals.plos.org/plosone/s/file?id=ba62/PLOSOne_formatting_sample_title_authors_affiliations.pdf

2. In the ethics statement in the Methods, you have specified that verbal consent was obtained. Please provide additional details regarding how this consent was documented and witnessed, and state whether this was approved by the IRB.

Line 221 we added: As per our Research Ethics Board ethics approval, the research coordinator reviewed the consent questions, and participants provided verbal consent at the outset of the interview.

We have uploaded our data to the Kaggle repository https://www.kaggle.com/datasets/sharonanderson/double-duty-caregivers/data

This is the correct information: The Northern Alberta Academic Family Medicine Fund R16P06 provided funding for this research.

“Thank you to the survey and interview participants who gave generously of their time.

Removed This study was funded by the Northern Alberta Family Medicine Fund”

“Northern Alberta Academic Family Medicine Fund”

We have included this statement in the cover letter: Thanks to the Northern Alberta Academic Family Medicine Fund R16P06 for this funding.

6. In the online submission form you indicate that your data is not available for proprietary reasons and have provided a contact point for accessing this data. Please note that your current contact point is a co-author on this manuscript. According to our Data Policy, the contact point must not be an author on the manuscript and must be an institutional contact, ideally not an individual. Please revise your data statement to a non-author institutional point of contact, such as a data access or ethics committee, and send this to us via return email. Please also include contact information for the third party organization, and please include the full citation of where the data can be found.

We have uploaded our data to the Kaggle repository https://www.kaggle.com/datasets/sharonanderson/double-duty-caregivers/data

7. We notice that your supplementary tables are included in the manuscript file. Please remove them and upload them with the file type 'Supporting Information'. Please ensure that each Supporting Information file has a legend listed in the manuscript after the references list.

We changed Supplementary files to Supporting file in the text and following the References. 

Checked reference list. There are no retracted papers in the list.

Reviewer #1: this manuscript is exceptionally well written and highly comprehensive. The authors related effectively a period that was detrimental to caregivers. their design is rigorous and reviewing it was a real delight.

I added minor comments to enhance the flow but generally speaking this manuscript has 5 stars.

Thank you so much. When the health system is so stressed by COVID-19 and the shortage of health providers, we were surprised when we didn’t find any research on double-duty caregivers. It came about because Dr. Lobchuk mentioned double duty oncology nurses cried in interviews when she asked if they were a family caregiver. Her study was not about double duty caregivers, but she talked about Dr. Ward Griffin retiring and the gap in knowledge about double duty caregivers in the pandemic. We applied to the Northern Alberta Family Medicine Fund (maximum funding $5000) and this is the result. 

Comments in attached manuscript

1. The question was “What about Canadians?” 

113 most American sons or daughters caring for their parents (86%) reported giving care was a

114 positive experience (42),

This reference was from a large US survey. I could not find similar research in Canada or the UK or we would have cited it. 

2. Line 372 “why don't you emphasize earlier in the text that you had hypotheses you did highlight them but it can be clearer as you are putting them in now”

We included the hypothesis in lines 178 to 185: “Building upon the theory put forth by Ward-Griffin and colleagues, [39,40] we hypothesized that greater expectations and blurred boundaries would be associated with higher levels of anxiety, while increased support, awareness of limits, boundary-setting, and making connections would be linked to lower anxiety. 

 We then hypothesized that caregiver health would be strongly associated with anxiety, and that age, gender, boundary blurring, expectations, support, awareness of limits, boundary-setting, connection-making, work hours, caregiving hours, and financial difficulties would also significantly contribute to caregiver anxiety. 

To make Hypothesis 2 clear we added in Lines 388-391 

Hypothesis 2 that caregiver health would be strongly associated with anxiety, and that age, gender, boundary blurring, expectations, support, awareness of limits, boundary-setting, connection-making, work hours, caregiving hours, and financial difficulties would also significantly contribute to caregiver anxiety was partially supported.

We added this text in Lines 396 to 398

Gender, familial and professional expectations, professional supports, making connections, caregiving hours and financial difficulties were not significant factors in anxiety. 

3. ???? on the word caree.

Line 411 we changed caree to care receiver 

4. Line 414 why don't you indicate all the themes that you did identify with the clear subthemes lined to the framework

As suggested, we have added starting in Lines 423:

The themes in the Double-Duty Caregiver domains include:

1. Expectations: “I’m overwhelmed and I’m exhausted” with subthemes in Self Expectations: “It’s just part of who I am.” and Familial expectations “Basically the perfect ingredients.”

2. Supports: “It’s like juggling balls” with subtheme on Making Connections “I had support because I worked in a hospital.”

3. The Caregiving Interface has three subthemes Making it work: “some sense of normality.” Working to manage “It’s a fine line” Living on the edge: “I was barely holding it together.”

5. Line 657 that's why it is important to make sure they are clear from the beginning of your manuscript

Thank you. We really appreciate your through review. 

6. Line 984 Funk(84, 85) add space here

Thank you, I have checked throughout the manuscript for spacing around the brackets. 

7. Line 689 performing medical tasks Add and nursing

We added nursing. 

8. Line 830 what about strengths and limitations of your study

Thank you, we added this section in lines 854-859

Strengths and Limitations

A strength of this study is the use of validated scales, including the Double-duty Caregiver Scale to examine double-duty caregivers’ experiences quantitatively and qualitatively. It highlights the anxiety related to the unique expectations from overlapping employment and family care roles. The cross-sectional snapshot is a limitation. While it captured increased stress at work and home, longitudinal research is needed. We know very little about how double duty caregiving affects work careers longitudinally.

Reviewer #2: Dear author

Your article concerning double-duty caregivers’ experiences during the COVID-19 pandemic is very interesting and relevant. The comments below are in support or clarification of some issues regarding the methodology.

Kind regards

Thank-you so much. We really appreciate the time and attention that reviewers put into peer reviews and how your review helped to make the paper stronger. 

Comments:

1. 202-204: How did you deal with the questionnaires that were not fully completed? Were they included/excluded?

We decided to include only those with complete data. Added to line 205 “We included only surveys with complete data.” 

2. 208: Can you explain why you chose 18 respondents for the follow-up interviews? + Can you provide the participants’ characteristics (how many of them were female/male/x)…?

Following Guest (1, 2), we assumed that we would have enough data after 12 to 15 interviews, but applied for funding for 18 just to make sure. In the future, we would follow Hennink’s (3) 2022 scoping review 9–17 interviews or 4–8 focus group discussions reached saturation. 

Re participant characteristics. We added this text starting in Line 209

All of our interview participants were female. We really wanted to recruit interview participants by the 3 caregiving Interface groups, but our research ethics board wanted us to collect email addresses on a survey separate from data. 

Added starting at Line 210 

Aligned with research advising that saturation is achieved with 12 to 15 interviews (71, 72), we sent emails to twenty respondents randomly selected from those who completed throughout the time the survey was active. For the first 18 respondents who replied to the email, a All of our interview participants were female. We wanted to recruit interview participants by the 3 caregiving Interface groups, but our research ethics board wanted us to collect email addresses on a survey separate from data. 

The research assistant arranged mutually agreeable interview times either via Zoom or with the first 18 respondents who provided contact information and who replied to the email. All were women. Because of the nature of our ethics (demographics not connected to identifying data for interview participation) we could not recruit interview participants demographically or theoretically. 

3. 258: Can you provide all four options of the Likert scale to the reader? + Why did you choose for a four-point Likert scale? For example, to avoid neutral answers?

We used the same four-point scale and scoring validated by the authors. The scale measures frequency of anxiety symptoms. We have added the four choices and the scoring.

Added in Line 266 1. Not at all, 2. Somewhat, 3. Moderately, 4. Very Much ranging from “not at all” to “very much.”

4. 292: How many of the participants answered the open-ended qualitative questions?

208 to 253 people answered the qualitative questions. 

5. 322-323: Can you explain why you chose the qualitative summative context analysis methods of Hsieh and Shannon?

 Often, we use Braun and Clarke’s thematic analysis, however because we were using the Double-duty Caregiver Scale theoretically, specifically looking for the themes related to the domains in Double-duty Caregiver Scale, we decided content analysis was the correct qualitative method in this study. 

6. 319-321: Did you obtain data saturation?

We achieved data saturation at 8 to 10 interviews. 

7. In general: Please place the references before the full stop that marks the end of the sentence.

Thank you. They are consistently before the full stop. 

1. Guest G, Bunce A, Johnson L. How Many Interviews Are Enough? An Experiment with Data Saturation and Variability. Field Methods. 2006;18(1):59-82.

2. Guest G, Namey E, Chen M. A simple method to assess and report thematic saturation in qualitative research. PLoS One. 2020;15(5):e0232076.

3. Hennink M, Kaiser BN. Sample sizes for saturation in qualitative research: A systematic review of empirical tests. Social Science & Medicine. 2022;292:114523.

---

## [Editor Report · Decision Letter 1]

29 Jan 2024

Double-Duty Caregivers Enduring COVID-19 Pandemic to Endemic: “It’s just wearing me down”

PONE-D-23-31596R1

Dear Dr. Anderson,

We’re pleased to inform you that your manuscript has been judged scientifically suitable for publication and will be formally accepted for publication once it meets all outstanding technical requirements.

Kind regards,

Stefaan Six, Ph.D.

Academic Editor

PLOS ONE

---

## [Editor Report · Acceptance letter]

4 Apr 2024

PONE-D-23-31596R1 

PLOS ONE

Dear Dr. Anderson, 

I'm pleased to inform you that your manuscript has been deemed suitable for publication in PLOS ONE. Congratulations! Your manuscript is now being handed over to our production team.

Kind regards, 

on behalf of

Dr. Stefaan Six 

Academic Editor

PLOS ONE